# What Does the Chinese Public Care About with Regard to Primary Care Physicians: Trustworthiness or Competence?

**DOI:** 10.3390/medicina55080455

**Published:** 2019-08-09

**Authors:** Egui Zhu, Yang Cao

**Affiliations:** 1Faculty of Education, Hubei University, Wuhan 430062, China; 2Clinical Epidemiology and Biostatistics, School of Medical Sciences, Örebro University, 70182 Örebro, Sweden

**Keywords:** public trust, primary care, hospital preference

## Abstract

*Background and Objective:* China has launched a series of reforms to enhance primary care. The aims of these reforms are to strengthen the functionality of primary care to encourage patients to use primary care. Patients’ trust in physicians is important in clinical medicine; however, little is known about how Chinese patients’ preferences relate to their trust in primary care physicians. This study’s objectives are to measure the Chinese public’s trust in primary care physicians and to characterize reasons of their preferences for health care. *Materials and Methods:* This quantitative study comprises a face-to-face survey with a convenience sample (*n* = 273) of people visiting community health centers or stations (CHCSs) in Wuhan, China. We measured the patients’ preferences for the different level of hospitals and their trust in physicians, as well as the reasons of the patients’ preferences, using a Chinese version of the Wake Forest Physician Trust Scale and other variables (such as demographics, health status, and hospital preference). *Results:* Approximately two thirds (68.6%) of the participants had experienced a mild or chronic disease in the year before the survey, but only 26.4% preferred to visit CHCSs in such cases. The negative factors related to this lack of preference are the physicians’ competence (odds ratio [OR] = 0.250), the medical equipment (OR = 0.301), and the popularity of hospitals (OR = 0.172). The positive factors were ease of access (OR = 2.218) and affordability (OR = 1.900). The participants expressed a moderate trust in physicians in CHCSs (score of 3.02 out of 5). There is no association between the patients’ trust and their hospital preference (*r* = 0.019, *p* = 0.859). Of the participants, 92 suggested that the physicians in CHCSs should improve in terms of their competence (*n* = 53), attitude (*n* = 35), and/or medical ethics (*n* = 16). *Conclusions:* This study’s results suggest that patients consider improving physicians’ competence to be more important and urgent than improving those physicians’ trustworthiness in terms of reconstructing Chinese primary care. Improving the physicians’ competence would not only reduce the barriers that patients experience regarding CHCSs, but would also increase their trust in the physicians.

## 1. Introduction

Primary care is the foundation of the health care delivery system to ensure equal access and reduced costs. The Chinese government launched a series of reforms to rebuild its underdeveloped primary care system during the market economic reform period [1,2]. For example, the Deepening Healthcare System Reform began in 2009, and 850 billion Chinese yuan (US$127 billion) was invested in improving primary care [3]. However, Wu et al. reported that the use of primary care was still limited because the public had little trust in the primary care system [4]. Based on the scope of the health care services provided, Chinese hospitals are divided into three levels (tertiary, secondary, and primary), with tertiary hospitals being the highest level. The primary care includes Chinese community health centers or stations (CHCSs), which are located in urban areas, and township health care centers, which are located in rural areas. Before the health care reform, most of the Chinese public could freely choose an affordable hospital to visit when they got sick [5]. However, health care became more expensive and more difficult to access when patients began frequenting high-level hospitals. In return, the patients’ discontent towards health care providers also increased, leading to mistrust between the patients and physicians [6].

The lack of public trust in China’s market-oriented health care system reform became a crisis [7]. Public trust is particularly critical in primary care, where physicians commonly lack competence, financial support, and equipment [8]. However, a study casted doubt on the anecdotal evidence regarding Chinese patients’ trust levels, showing that patients had a high level of trust (80%) in the physicians at public hospitals (including all three levels of health care services provided) in Beijing [9]. Also, Dong et al. showed that patients had a medium level of trust (54%) in physicians at the tertiary hospitals in Shanghai [10,11]. Beijing and Shanghai are the biggest and most developed cities in China and have better health care delivery systems than other places. However, there is a lack of evidence about public trust in primary care physicians in medium to smaller cities in China, and whether the trust in primary care physicians affects preferences for the choice of primary, secondary, or tertiary hospitals.

The measures of patients’ trust can differ for diverse reasons [12]. However, it has been shown that patients’ trust in primary care physicians has the following core dimensions of trust: Interpersonal and technical competence, moral comportment, and vigilance [13].Three measurement tools have been shown to have high reliability. They are: The Primary Care Assessment Survey [14], which does not focus on trust; the Trust in Physicians Scale [15], which does not measure competence; and the Wake Forest Physician Trust Scale (WFPTS), a 10-item scale which measures patients’ trust across four dimensions—competence, fidelity, honesty, and global trust [16,17]. Developed especially for primary care, the WFPTS has also been used to measure patients’ trust in specialist physicians in various countries, including China [11,18]. The results of previous studies indicate that the WFPTS has high reliability (Cronbach’s α = 0.83–0.92). In general, the WFPTS results show that patients have high trust in, and satisfaction with, physicians, as well as good behavioral consequences. Therefore, we selected WFPTS as the suitable measure to assess patients’ trust in China.

To investigate the reasons of preferences in primary care, the public trust in primary care physicians in China, and factors that influence preference and trust, we conducted a cross-sectional survey and assessed the following questions:
What factors affect patients’ choices of primary care?What level of trust do the public have in primary care physicians?Does trust in primary care physicians affect patients’ choices of primary care?What do patients expect of primary care physicians?

## 2. Materials and Methods

We used the Chinese version of the WFPTS (C-WFPTS) in this study to measure public trust in primary care physicians in Wuhan City, Hubei Province, China. Although the C-WFPTS has been tested in some tertiary hospitals in China (Cronbach’s α = 0.83), the new translations that we used in this study have not been tested in CHCSs. Therefore, we conducted a pilot study to test the acceptability and reliability of the survey instrument before the formal survey.

### 2.1. Study Setting

Wuhan is one of the ten cities that have participated in the reforms of the Chinese general practitioner service mode since 2012. The population of Wuhan was 10.12 million in 2012. Wuhan had planned to build more health care providers, particularly primary care, as well as to set a national model for providing quality basic health care services. There are 450 CHCSs in Wuhan [19], and most of them are located in the city’s seven main districts. We conducted this study at seven CHCSs by randomly selecting one CHCS from each of the seven main districts in Wuhan.

### 2.2. Sampling of Study Participants

We used the convenience-sampling method for the survey. Because this study is explorative, we did not define any hypotheses or make a sample-size calculation. We conducted, in total, 300 face-to-face questionnaire interviews with the visitors whom we met in the reception areas of the seven selected CHCSs in a single day. The CHCS visitors completed the survey voluntarily without remuneration. They were allowed to return the questionnaire later by post if they wanted. The visitors were patients, family members, or friends who accompanied the patients to the CHCS. We did not distinguish between the three kinds of visitors in our analysis. The survey was anonymous but all the participants read the consent form and gave oral consent face-to-face before they filled out the questionnaire. The Ethics Committee of Tongji Medical College at Huazhong University of Science and Technology approved this study (approval number: S545,2012-11-28).

### 2.3. Study Instrument

#### 2.3.1. Trustworthiness of Physicians

Patients’ trust was assessed using two versions of the C-WFPTS in a pilot survey and a formal survey, respectively.

In the pilot survey, we used the version of the C-WFPTS that Dong et al. translated and adapted [10]. The pilot study was used to evaluate and improve the feasibility and reliability of the questionnaire used for the formal survey. EZ, one of the authors, conducted the face-to-face pilot survey for the CHCS visitors in May 2012 to determine the feasibility of the survey. Both EZ and a psychologist, Li Nv (NL), checked the responses. During the data collection, we marked the items that the respondents found hard to understand and considered how to rephrase them.

In the formal survey, we rephrased the items that the pilot participants had difficulty to understand. We found that some of these items had been incorrectly translated from the original WFPTS. If an item had different meanings in the C-WFPTS and in the WFPTS, we retranslated the original WFPTS item into Chinese to better fit with the original WFPTS. 

Both versions of the questionnaire include 10 items that are scored on a 5-point-Likert scale (from 1, totally disagree, to 5, totally agree). The differences between the pilot version and the formal version (in both Chinese and English) are given in the Appendix A.

#### 2.3.2. Socio-Demographic Variables, Health Status, and Hospital Preference

To get more information on the patients’ views of CHCSs, we added five structured questions on the patients’ basic health situation, hospital choice, and the reason for the choice. The first four items relate to the participants’ demographic characteristics: Sex, age, education, and income. The last question is open-ended and asks about the patients’ opinions regarding the issue that physicians in CHCSs most urgently need to improve upon.

### 2.4. Data Collection

In the pilot study, 200 subjects were approached and 166 returned the questionnaire. After the pilot study, NL trained seven psychology undergraduate students from Hubei University to conduct the questionnaire interview for the formal study. In November 2012, the trained students distributed 300 formal questionnaires to participants whom they met in person at the seven selected CHCSs. Two students worked together to transcribe the data from the paper questionnaires into an electric form. We reversed the scores of the negative items in the C-WFPTS so that the scale was from 1 (totally agree) to 5 (totally disagree). EZ checked the accuracy of the electronically transcribed data against the responses in the original paper questionnaire.

### 2.5. Data Analysis

In the pilot study, we used item-to-total correlations and a response distribution to check the items’ reliability. We then reexamined any items with an item-to-total correlation lower than 0.3. We used the Cronbach’s α and mean inter-item correlation coefficients to evaluate the reliability of the C-WFPTS; a high α indicates high reliability [20]. The minimum acceptable α value was 0.7, and we considered a value greater than 0.8 to be good [21,22]. We used an optimal mean range for the inter-item correlation coefficient of between 0.2 to 0.4 and an acceptable range of between 0.1 and 0.5 [18].

In the formal study, we summarized the participants’ characteristics, their preferences for the hospitals, and the frequencies of the reasons related to the preferences. We used Pearson’s chi-squared test and odds ratios (ORs) to identify factors that potentially influenced the patients’ hospital preferences. We used principal component (PC) analysis and the item-scale correlations to analyze the C-WFPTS. We compared the differences in the PCs of the patients’ trust using the analysis of variance (ANOVA) for repeated-measures. We assessed the magnitude of the relationship between the patients’ trust and the participants’ demographic and socioeconomic characteristics, health status, and hospital preferences using the point-biserial correlation test, bivariate correlation analysis, and Pearson’s correlation coefficient. We used thematic analysis to summarize the results for the open-ended question. All the analyses were conducted using IBM SPSS statistical software (version 24; IBM Corp., Armonk, NY, USA).

## 3. Results

### 3.1. Participants’ Demographic and Socioeconomic Characteristics 

We received 274 completed questionnaires in the formal survey, resulting in a response rate of 91.3%. After excluding the participants who had missing values on more than 5% of the total items (for both the C-WFPTS and the demographic and health status questions) and who indicated age less than 18 years in the questionnaire, 229 participants remained. The participants’ demographic and socioeconomic details are shown in Table 1.

### 3.2. Hospital Preferences

Of the 229 participants with complete information, only 26.2% preferred CHCSs; another 24.5% opted for tertiary hospitals, and 21.8% chose secondary hospitals. The rest engaged in self-medicine (19.2%) or went to private clinics (8.3%).

#### 3.2.1. Health Situations and Hospital Preferences

During the year leading up to the survey, 68.6% of the participants had health problems, but most of the problems were mild or chronic. Only 3.5% of the participants had a serious disease that could not be treated at a CHCS under the new Chinese health care rules. The participants’ health situations and their hospital preferences are shown in Figure 1.

#### 3.2.2. Reasons for Choosing a Hospital

The reasons that the participants chose the hospitals are as follows: Ease of access (*n* = 100, 43.7%), physicians’ competence (*n* = 72, 31.4%), acceptable price (*n* = 55, 24.0%), medical equipment (*n* = 44, 19.2%), physicians’ attitude (*n* = 40, 17.5%), insurance-payment policy (*n* = 34, 14.8%), popularity (*n* = 30, 13.1%), and other (*n* = 3, 1.3%). The Pearson’s chi-squared test results indicate that the following factors have a statistically significant association with preference for a tertiary hospital or a CHCS: Physicians’ competence, accessibility, popularity, price of treatment, and medical equipment. As shown in Table 2, compared to those who did not consider accessibility, those who did were more likely to visit a CHCS (OR = 2.218) or a private clinic (OR = 4.005), but were less likely to visit a tertiary hospital (OR = 0.198). Compared to those who did not consider the physicians’ competence, those who did were more likely to visit a tertiary hospital (OR = 6.081) but less likely to visit a CHCS (OR = 0.250) or to self-medicate (OR = 0.359). Compared to those who did not consider price, those who did were more likely to visit a private clinic (OR = 3.189) but less likely to visit a tertiary hospital (OR = 0.301). Compared to those who did not consider medical equipment, those who did were more likely to visit a tertiary hospital (OR = 3.426) but were less likely to visit a CHCS (OR = 0.301). Compared to those who did not consider popularity, those who did were more likely to visit a tertiary hospital (OR = 3.250) but less likely to visit a CHCS (OR = 0.172). The patients who visited secondary hospitals were more likely to consider physicians’ attitudes (OR = 2.274), which is the only statistically significant factor with regard to choosing secondary hospitals. There is no association between the insurance-payment policy and the patients‘ hospital preferences.

### 3.3. Public Trust in Physicians

#### 3.3.1. Reliability and Validity of the C-WFPTS

The pilot study’s results show that the C-WFPTS has acceptable reliability (Cronbach’s α = 0.794, *p* < 0.001, α > 0.70). Two items (Q10 and Q12) had item-scale correlation coefficients (0.134 and 0.156, respectively) lower than the acceptable threshold 0.3. The remaining items had correlation coefficients greater than 0.421. The participants reported that one item (Q17) that we had adapted to fit the Chinese context was actually impossible for physicians to achieve. In addition, the participants rated two items (Q13 and Q17) as being difficult to understand.

After rephrasing the difficult-to-understand items to make them closer in meaning to the items in the original WFPTS, we found that the C-WFPTS had a Cronbach’s α of 0.821 (*p* < 0.001) and a mean inter-item correlation coefficient of 0.308. The item-scale correlation coefficients of the two lower-scoring items (Q10 and Q12) increased (0.365 and 0.286), but the coefficient for Q12 was still unacceptable. The coefficients for the rest of the items of the survey were greater than 0.481.

#### 3.3.2. Principal Components

The survey’s Kaiser-Meyer-Olkin value was 0.832 (*p* = 0.000). We identified two PCs with eigenvalue greater than 1 that could explain 56.97% of the total variance. The first PC includes all the positive items, and the second one includes the three negative items (Q10–12). However, we observed a visually significant decrease after adding the third PC to the scree plot (see Figure 2). In the three-PC model, we extracted two items (Q13 and Q14) from the first PC. When using four PCs, we extracted Q16 from the first PC. However, Q10 and Q16 both remained in two PCs. Thus, we used the three-PC analysis to describe the items, which broke the items into three categories: Competence (Q10–12), attitude (Q13 and Q14), and global trust (Q15–19). 

#### 3.3.3. Public Trust

A high item score (closer to 5) indicates high trust for these items. In this study, the C-WFPTS questions have a mean score of 3.02 and a median of 3.0, and the questions’ means have a range of 1.0–4.6, which indicates a moderate level of trust in physicians at the CHCSs. The scores of the three PCs conform to the assumptions of normality and sphericity, and the repeated-measures ANOVA shows a statistically significant difference (*p* < 0.001) in the mean scores of the following PCs: Attitude (mean = 3.18, standard deviation = 0.79), competence (mean = 2.95, standard deviation = 0.82), and global trust (mean = 2.99, standard deviation = 0.72). 

#### 3.3.4. Associations with Patients’ Trust

The results of the point-biserial correlation test show that the patients’ trust is not associated with their preference for CHCSs (*r* = 0.019, *p* = 0.859). The results of the bivariate correlation analysis (see Table 3) indicate that the patients’ trust is inversely associated with their education (*r* = −0.295, *p* < 0.001) but is not correlated with the other variables (age, income, payment, and health situation).

### 3.4. CHCS Physicians’ Competence

#### 3.4.1. Physicians’ Competence and Patients’ Trust

The items about trust in competence are the most important, as most participants reported that competence could affect their trust in physicians. In particular, the participants noted that Q11 (“[The CHCS doctor’s] medical skills are not as good as they should be”) was the most important item in terms of the patients’ trust. Overall, 25.6%, 18.5%, and 11.3% of the participants ranked Q11 as the first-, second-, and third-most important item associated with trust, respectively.

#### 3.4.2. Participants’ Expectations

The open-ended question was the last question in the questionnaire, and the participants could freely give their answers. We received responses to the question from 85 participants (37.1%), and extracted four themes from the responses. Most of the participants (51) reported that CHCS physicians should improve their competence; 28 said that physicians should have better attitudes; 16 said that physicians should improve their medical ethics; and 11 hoped that the price of health care services could be reduced.

We quote some examples of the answers below:
“(They need to) improve their diagnosis ability and avoid the mistake that might cause patient’s loss and health problems.”(Participant 226)
“(They need to) improve their medical competence through continuing medical education.”(Participant 015)
“(They need to improve) medical care competence and attitude towards patients.”(Participant 142)
“(They should) make treatment decision from the patients’ perspective and avoid the unnecessary diagnostic tests.”(Participant 141)

## 4. Discussion

This study shows that Chinese patients have varied facility preferences, and CHCSs are not the preferred health care providers, even among those who do not have a serious disease. These findings are consistent with those from a study on Chinese physicians’ opinions [4]. We found that the C-WFPTS can be used to measure the public’s trust in primary care physicians in China (Cronbach’s α = 0.813, *p* < 0.001), which follows the suggestion of Dong et al. for using this survey widely in various Chinese health care institutions [11].

### 4.1. Comparison with Other Studies

Unlike Dong et al. [10], we found that only the patients’ education level was associated with their trust in physicians. Also, although a previous study designed four dimensions [16], we found a two-dimensional structure, just as other researchers have reported [11,18,20]. Furthermore, we found that trust has three dimensions: Competence, attitude (or morality), and global trust. We extracted the attitude or morality dimension from global trust using a three-PC analysis, as suggested in an integrative review [13]. Hall et al. reminded other scholars to pay attention to the failed differentiation of competence dimensions within trust [20]. They explained this failure as resulting from either the participants’ overall high trust scores or their difficulty in differentiating between technical competence and communication competence [18,20]. This failure could aslo be due to the difference between the participants’ understanding of the items and the tool designer’s expectations. For example, extremely thorough and careful service could be considered as an aspect of attitude rather than an aspect of competence. To use multiple dimensions when measuring trust, we need to improve the items’ ability to distinguish in order to fit with the original four dimensions.

The results of this study indicate that the Chinese patients’ trust in primary care physicians is greater than their trust in specialist physicians [11]. However, compared to the results of studies where researchers used the same tool to evaluate patients’ trust in physicians, these trust levels are lower than those reported in other geographical areas (which had means ranging from 3.88 to 4.3) [18,20,23]. This low level of trust in Chinese physicians could be related to the market-oriented reforms. In a cross-national comparison study, scholars reported that trust in physicians had deteriorated since this commoditization [24]. High-quality, trusted, professional physicians have faced challenges resulting from the turbulent changes in China’s health care system [25]. Scholars have reported that trust in physicians has declined since the implementation of the market-oriented health care reforms [26]. This trust is hard to rebuild, particularly if the health care reforms fail to restrict the providers’ pursuit of profit [4]. The low level of trust in Chinese physicians could also be attributed to insufficient communication between the patients and physicians because patients usually visit many physicians within a short time, and their trust in physicians is also related to the continuity of the patient–physician relationship [27].

### 4.2. Implications for Practice

The level of trust between physicians and patients may affect the effectiveness of treatments [28]. Patients’ trust in physicians is essential to ensure the continuity of the patient–physician relationship in patient-centered care [12,27,29]. Moreover, high-level trust can improve treatment outcomes by smoothing communication between patients and physicians, improving patients’ adherence to treatment, and reducing patients’ fear [30]. Patients’ trust can also affect physicians’ medical decision-making by, for example, reducing the unnecessary prescription of medication [31,32]. In this study, we found that patients did not have a high degree of trust in Chinese primary care physicians; however, we cannot conclude that the low level of preference for CHCSs is due to the distrust in the physicians at those centers. On the one hand, we did not find an association between trust in physicians and hospital preferences. On the other hand, we found a moderate level of trust, with a mean trust score (3.02) that is slightly higher than the one (2.69) found for tertiary hospitals in Shanghai, China [11].

Our results suggest that negative factors associated with CHCSs include physicians’ competence, medical equipment, and popularity, whereas the positive factors are the ease of access, as well as affordability. These factors, together, could lead to public overuse of high-level hospitals and underuse of primary care. In one study, researchers showed that Chinese patients’ preferences for hospitals were highly related to their trust levels, as they distrusted primary care but not high-level hospitals [33]. It is thus necessary to reduce the negative effects of these factors in CHCSs so that they can provide the most public health services. The Chinese government has tried to launch primary care reforms since 1997 [34]. It has invested increasing amounts in these reforms, from $2.8 billion in 2008 to $20.3 billion in 2015 [35]. Among the five priorities of these Chinese health reforms, the one with the slowest progress is the development of a primary health care service [36]. The policy of encouraging patients to use CHCSs has failed in part because of the low quality of the CHCSs in the pilot cities [37].

Researchers have reported that physicians’ incompetence could limit the utilization of CHCSs, and thus hinder the efforts to improve the access, quality, and efficiency of public health services [34]. The results of this study indicate that the public trust in physicians’ competence is one of the key factors in attracting patients to visit CHCSs; therefore, improving physicians’ competence is important to increase the utilization of CHCSs. The public trust in physicians’ competence is slightly lower than in physicians’ attitude and global trust. There are two potential reasons for this difference. First, compared with the specialists at high-level hospitals, primary care physicians have relatively lower education [2]. Second, most Chinese primary care physicians lack the general practitioner (GP) training provided in Western countries [38]. GP education was introduced to China in the late 1980s but was not truly initiated until 1999 [39]. The Chinese national GP undergraduate education program was not established until 2010; this program is mainly designed for training GPs to work in rural areas [40].

### 4.3. Limitations

We realize that this study has several limitations. First, although the seven CHCSs are a random representative sample of the CHCSs in Wuhan, the convenience sample of the CHCSs visitors that we interviewed limits the generalization of the study’s findings. Second, we did not distinguish between the patients and accompanying family members and friends, which may have led to selection bias. Third, although WFPTS has high reliability and validity, it is difficult for common people to assess the competence of physicians when they visit CHCSs, which may have led to measurement bias. These remind us to interpret our findings cautiously.

## 5. Conclusions

This study’s results indicate that, in China, primary care physicians’ competence is important to patients. It influences not only the patients’ trust, but also their preferences for facilities. Few of the surveyed participants thought that Chinese primary care physicians had sufficient competence. In addition, many of the patients with common or chronic diseases preferred tertiary or secondary hospitals.

## Figures and Tables

**Figure 1 medicina-55-00455-f001:**
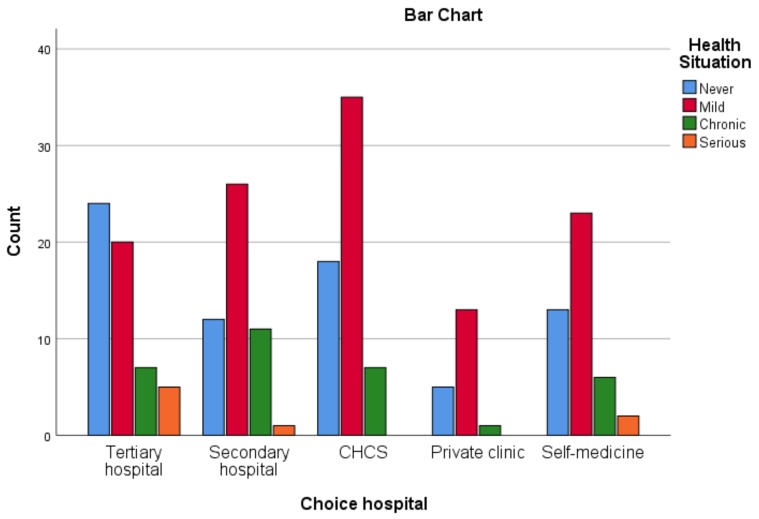
Participants’ health situations and hospital preferences.

**Figure 2 medicina-55-00455-f002:**
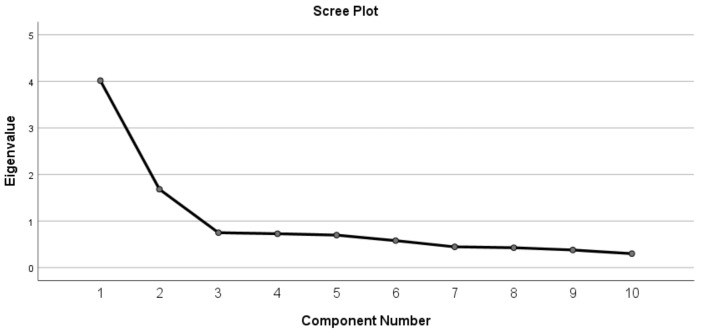
Screen plot of the factors in the C-WFPTS (*N* = 229).

**Table 1 medicina-55-00455-t001:** Demographic and socioeconomic characteristics of the participants (*N* = 229).

Variable		*N*	%
Gender	Male	111	48.5
	Female	118	51.5
Age	18–39	122	53.3
	40–59	77	33.6
	≥60	30	13.1
Education	≤Middle school	33	14.4
	High school or technical school	61	26.6
	Some college	39	17.9
	Undergraduate degree	78	34.1
	≥Graduate degree	18	7.9
Monthly income	≤1500 yuan	61	26.6
	1501–4000 yuan	130	56.8
	4001–7000 yuan	33	14.4
	7000–10,000 yuan	4	1.7
	10,001–30,000 yuan	1	0.4
Payment method	Urban-Resident Basic Medical Insurance	58	25.3
	Urban-Employee Basic Medical Insurance	71	31.0
	Government-Funded Medical Insurance	32	14.0
	New Rural Medical Scheme	17	7.4
	Commercial health insurance	3	1.3
	Self-payment	42	18.3
	Others	6	2.6

**Table 2 medicina-55-00455-t002:** Odds ratios (95% confidence intervals) of hospital preferences (*N* = 228; 1 case missing).

	Tertiary	Secondary	CHCS	Private Clinic	Self-Medication
**Accessibility**	0.198 (0.094–0.418) ***	0.734 (0.386–1.396)	2.218 (1.218–4.042) **	4.005 (1.391–11.532) **	1.437 (0.739–2.795)
**Competence**	6.081 (3.169–11.671) ***	1.292 (0.668–2.499)	0.250 (0.112–0.560) ***	0.553 (0.177–1.729)	0.359 (0.151–0.852) *
**Price**	0.301 (0.121–0.748) **	0.531 (0.232–1.213)	1.900 (0.989–3.651)	3.189 (1.223–8.314) *	1.278 (0.605–2.702)
**Equipment**	3.426 (1.708–6.873) ***	1.666 (0.794–3.486)	0.301 (0.113–0.804) *	0.468 (0.104–2.104)	0.372 (0.125–1–102)
**Attitude**	0.729 (0.314–1.691)	2.274 (1.08–4.786) *	0.540 (0.225–1.297)	0.872 (0.242–3.145)	1.093 (0.464–2.576)
**Payment**	1.341 (0.597–3.009)	1.604 (0.710–3.626)	1.009 (0.442–2.306)	0.651 (0.143–2.954)	0.373 (0.108–1.281)
**Popularity**	3.250 (1.469–7.190) **	1.641 (0.699–3.852)	0.172 (0.40–0.747) **	0.761 (0.167–3.471)	0.439 (0.127–1.520)

* *p* ≤ 0.05; ** *p* ≤ 0.01; *** *p* ≤ 0.001.

**Table 3 medicina-55-00455-t003:** Analysis of associations for patients’ trust.

Variable	Patients’ Trust	
	Pearson Correlation Coefficient	*p*
Age	0.072	0.278
Education	−0.295 **	<0.001
Income	−0.078	0.238
Payment	−0.0030	0.655
Health situation	−0.018	0.782

** *p* < 0.001.

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
