# Peer review of "What Does the Chinese Public Care About with Regard to Primary Care Physicians: Trustworthiness or Competence?"

_medicina, 2019, doi:10.3390/medicina55080455_

Round 1

Reviewer 1 Report

Esteemed Authors,

It has been a great honor, as well as a pleasantly challenging activity, to review the article entitled What does the Chinese public care about with regard to primary care physicians: trustworthiness or competence?”.

In order to achieve a high level of health protection for the public and to guarantee their right to information, the main actors in medical care it should be ensured that public is appropriately informed as regards the care services. Usually, the patients’ choices can be influenced by many factors: the major factors who influence the patients' choices, are represented inter alia, health, economic, social and ethical considerations.

The perception of the public about the phenomenon of care and about the doctors directly involved in this phenomenon – primary care physicians, in this case – is very important for choosing a particular option.

Within the above-mentioned article, the authors have chosen to provide a detailed picture of the perception of Chinese public care, perception established directly by patients, as consumers and beneficiaries of health care services.

The article approaches a topic that is highly significant to the Chinese public as well as to public health systems and their evolution, in order to ensure improvement, especially when it comes to the interaction between these systems and public policies in the considered area.

The article is structured following the classic model for this type of article – original research article - comprising 5 parts: introduction; materials and methods; results; discussion and conclusions. The 5 major components of the article are balanced dimension-wise and are presented coherently and logically, tightly linked to one another.

All materials and methods are specified and described adequately. All iconographic materials – 3 tables and 2 figures - were given accurate descriptions, the results were described in great detail and the conclusions are adequate.

The documentation is adequate, and all the authors are cited in the text of the paper. The provided scientific results are exact and precise. The goal of the conducted research is well specified and delineated. The working protocol is appropriate, and the used analysis methods are correlated with the proposed objectives.

Nevertheless, the detailed analysis of the paper has also highlighted some aspects that require revision, as follows below:

All references are cited in the text, the bibliography is relevant, but presents some minor lacks when it comes to citations and/or mentions. In order to clarify some aspects, I would suggest that the authors write the bibliography evenly: for example, journal papers require either the complete journal name, or the JCR abbreviation (in the case of ISI indexed or rated journals), or the ISO abbreviation (for BDI indexed journals); moreover, for journals, I suggest that the volume, number, and pages (as the case requires) be mentioned. I would also recommend that greater attention be paid when it comes to chapters from books and that the number of pages, publishing house and other identification elements (link, Digital Object Identifier – DOI, etc.) be mentioned, regardless of the reference type.

For example: page 11, lines 374-375, number 5 in the references list: Eggleston Karen, Ling L., Qingyue M., Lindelow M., Wagstaff A. (2008). Health service delivery in China: a literature review. Health Economics (or JCR Abbreviation – Health Econ.), 17, 2, 149-165; DOI: https://doi.org/10.1002/hec.1306.

Another example: page 11, lines 376-378, number 6 in the references list: Tucker J.D., Cheng Y., Wong Bonnie, Gong N., Nie J.B., Zhu W., McLaughlin Megan, Xie R., Deng Y., Huang M., Wong W.C.W., Lan P., Liu H., Miao W., Kleinman A. (2015). Patient physician mistrust and violence against physicians in Guangdong Province, China: a qualitative study. BMJ Open (or JCR Abbreviation – BMJ Open), 5, 1-10:e008221; DOI:10.1136/bmjopen-2015-008221.

            Another example: page 12, lines 394-396, number 14 in the references list: Safran Dana Gelb, Kosinski M., Tarlov A., Rogers W., Taira Deborah, Lieberman Naomi, Ware J.E. (1998). The Primary Care Assessment Survey: Tests of Data Quality and Measurement Performance. Medical Care (or JCR Abbreviation – Med. Care), 36, 5, 728-739.

Another example: page 12, lines 412-413, number 21 in the references list: Fraenkel J.R., Wallen N.E., Hyun Helen (2012). How to Design and Evaluate Research in Education. Eighth Edition. The McGraw-Hill Companies, Inc., New York.

I would advise the authors to be more careful with regard to the bibliography: it is preferred that the cited authors be mentioned in alphabetical order, and references without specified authors be mentioned at the end of the list of references, in chronological order. I also recommend using a single system not only in citations but also when it comes to the journals. I am referring here mainly to mentioning the following elements for each article consulted: journal, volume, issue and pages (the DOI may also be mentioned, should the author so desire, but the basic descriptive elements are the previously mentioned ones).

The obtained results are interpreted correctly, and their practical value is visible.

As for the paper grammar, the article is generally very well written: only a few shortcomings in the grammar of the text can be mentioned, as follows:

Page 6 line 218 – replace ,, As the eigenvalue’’ with ,,Considering the fact that the eigenvalue’’;

Page 9, line 301 – replace ,, between patients’’ with ,,between the patients’’;

Page 10, line 339 - replace ,, between patients’’ with ,,between the patients’’.

As a general conclusion regarding the grammar, the text does not contains other mistakes that need to be corrected. As for the editing (writing) is concerned, the text should be checked once again carefully.

The article itself, like any other article, has certain improvable aspects. By these aspects, I mean the major constituting parts of the article, but also some elements that are related to details or writing. However, the article as a whole, despite not having a very high degree of originality, can be considered interesting for academic staff, for researchers in the field and even for the wide public.

Together with other positive elements, the scientific relevance and quality of the presentation will surely make the article attractive to a wide audience, and especially to authors interested in the fields of medical care, and public health.

Provided that the authors revise their paper and improve on the elements mentioned above, the paper may be published in the Medicina.

            Best Regards,

            Reviewer

Author Response

 we thank reviewer' recognition ou our manuscript. And we than reviewers constructive comments and suggestions. We have adapted the manuscipt  according to the insightful comments. Please see the attachment. 

Reviewer 2 Report

In their manuscript, Zhu and Cao describe their study of patient attitudes towards physician trustworthiness and competence in primary care facilities in Wuhan, China. While the result from this study will be very important information for improving medical care, the study itself is confusing and possibly fatally biased and much more explanation and description of the methodology as well as the results are needed.

1)   The overall manuscript could use a bit of editing for English language, grammar, and syntax.

2)   In the Introduction, please consider putting the Chinese community health centers or stations in context compared to other medical options in China. How do these centers/stations compare to primary medical facilities in other places mentioned in the literature (ie: Beijing, Shanghai, etc…) and where do they fit into this system? How many people use these facilities compared to other available options? This information is available in the methodology, but would likely be better served to put context to the study if used in the introduction.

3)   Lines 94-96: Are the selected studies representative of population, etc..? Was one CHCS selected per district?

4)   Lines 99-101: Did the authors meet 300 individuals per CHCS or 300 total? Why were questionnaires not conducted on site?

5)   It is very confusing that the authors mention both a pilot study and a formal study; however, only seem to describe the pilot survey? The difference between which study was conducted when, how many people were involved, and how the answers were used needs to be explained in much more detail. In addition, with a pilot survey conducted first, there should be a both a hypothesis and a powered sample-size calculation conducted for the formal survey – this is a potential fatal flaw to the study unless better explained.

6)   Was consent completed face-to-face? Was incentive offered for completion of the survey?

7)   Lines 159-161: were these questionnaires not given in person (as suggested by the mention the participants were met “face-to-face”?). If so, how were almost 10% of surveys incomplete?

8)   Lines 162-163: How could these participants under the age of 18 have been included in the study if they were under the age of 18? They, technically, could not provide their own consent so should not have been approached in the first place.

9)   How does the demographics of the population compare to the population demographics of the community as a whole?

10)                  Lines 167-169: if individuals preferred other means of healthcare, then why were they at the CHCS the day of the study? How does this contribute to bias in the study?

11)                   Lines 247-252: How qualified are the participants to judge the competence of physicians? Ie: what context do they have to say that the “…medical skills are not as good as they should be…”? Making assumptions based on the expectations of individuals who have no medical expertise seems to generate a lot of bias in this study. This is mentioned in the limitations, but, as it is a major finding in the study, may need to be reconsidered in the analysis as it cannot really be proven that the physicians are incompetent based on responses from individuals who cannot truly assess the physicians competence. Please consider how to address this point.

12)                   Similar to the last statement, was any qualification given for the patients having distrust in attitude or ethics? Was any context given for these statements? Again, these seem like highly individualized, contextually biased questions.

13)                   Please expand the limitations and biases discussed in section 4.3.

Author Response

We thank the reviewer for reviewing our manuscript, and giving insightful comments and constructive suggestions. We have adapted the manuscript according to the comments and give point-to point response. please see the attachment. 

Round 2

Reviewer 2 Report

The authors have adequately addressed my previous concerns. Thank you.